# Phenotypic and Molecular Study of Multidrug-Resistant *Escherichia coli* Isolates Expressing Diverse Resistance and Virulence Genes from Broilers in Tunisia

**DOI:** 10.3390/antibiotics14090931

**Published:** 2025-09-15

**Authors:** Ghassan Tayh, Fatma Nsibi, Khaled Abdallah, Omar Abbes, Ismail Fliss, Lilia Messadi

**Affiliations:** 1Department of Microbiology and Immunology, National School of Veterinary Medicine, University of Manouba, LR16AGR01, Sidi Thabet, Ariana 2020, Tunisia; fatma.nsibi@yahoo.com (F.N.); abdallah.khaled1993@gmail.com (K.A.); 2DICK Company, Poulina Group Holding, Ben Arous 2034, Tunisia; abbes.omar@eddick.com.tn; 3Department of Food Sciences and Nutrition, University of Laval, Québec City, QC G1K 7P4, Canada; ismail.fliss@fsaa.ulaval.ca

**Keywords:** one health concern, virulence factors, antibiotic resistance, carbapenemase, extended-spectrum β-lactamases (ESBL), integrons

## Abstract

This study investigated the molecular and phenotypic characteristics of antimicrobial resistance in *Escherichia coli* isolates recovered from the ceca of healthy broilers in Tunisia. A total of 111 *E. coli* isolates were obtained from chicken samples collected at slaughterhouses and cultured on cefotaxime-supplemented MacConkey agar. All isolates exhibited a multidrug-resistant (MDR) phenotype, and 72.1% were confirmed as extended-spectrum β-lactamase (ESBL) producers. The most frequent β-lactamase gene was *bl*a_CTX-M-G1_, followed by *bla*_TEM_ and *bla*_SHV_. Carbapenem resistance genes (*bla*_OXA-48_ and *bla*_IMP_) were detected in 12.6% and 6.3% of isolates, respectively, while six isolates harbored the colistin resistance gene *mcr-1*. Among the tested virulence genes, *fimH*, *traT*, and *iutA* were the most prevalent, detected in over 70% of isolates. Class 1 integrons were present in 83% of isolates, and class 2 integrons in 39.6%, with gene cassettes encoding resistance to trimethoprim (*dfrA*) and streptomycin (*aadA*). These findings highlight the widespread presence of MDR and ESBL-producing *E. coli* strains with virulence traits and integrons in poultry, underscoring the risk of transmission to humans. This study provides essential data supporting the implementation of integrated surveillance strategies in line with the One Health approach.

## 1. Introduction

*Escherichia coli* belongs to the commensal microbiota that exists as harmless bacteria in the intestine of humans and warm-blooded animals [1]. However, many pathogenic strains can cause intestinal or extraintestinal infections in humans and animals [2]. The pathogenic *E. coli* strains are categorized into intestinal pathogenic isolates (InPEC) for diarrheagenic *E. coli* (DEC) that cause gastroenteritis and extraintestinal pathogenic *E. coli* (ExPEC) that cause urinary tract infection (UTI), meningitis, and pneumonia [2].

The capacity of these strains to cause infections is associated with several virulence determinants such as adhesins, hemolysin, fimbriae, and others, enhancing the bacterial survival in several conditions and potentiating the bacterial pathogenesis [3]. In bacterial pathogens, these factors are encoded by specific genes found on the chromosome or within mobile genetic elements, such as plasmids or transposons [4]. ExPEC strains possess various virulence factors that enhance their ability to infect the host by facilitating adhesion, iron acquisition, invasion, and toxin production. Adhesion molecules such as *fimH*, *BfpA*, *papGIII*, *sfa/focDE*, and *eae* promote bacterial attachment to host tissues, while the iron acquisition gene iutA aids in survival by securing essential nutrients. The invasion-associated gene *traT* contributes to immune evasion, and toxins, including *stx-1*, *stx-2*, *ehxA*, and *cdt3*, play a crucial role in pathogenicity. These factors collectively enable ExPEC to establish and sustain infections [5,6].

*E. coli* strains can be classified into seven phylogenetic groups (A, B1, B2, C, D, E, and F, and the eighth is the *Escherichia* cryptic clade I) by employing a multiplex PCR assay that was developed by Clermont et al. [7]. Different studies revealed that ExPEC strains belong to phylogroups B2 and D, while commensal and diarrhoeagenic strains belong to other phylogroups [8].

Avian pathogenic *Escherichia coli* (APEC) causes extraintestinal infections in poultry, including chickens, turkeys, ducks, and other avian species, leading to systemic colibacillosis characterized by common postmortem lesions. Mortality rates in young chickens can reach up to 53.5% [9]. Avian colibacillosis is considered one of the major causes of economic losses in the poultry sector worldwide. In the United States, more than USD 40 million is expected to be lost in the broiler sector each year as a result of carcass condemnation [10]. APEC shares significant genetic similarity with human extraintestinal pathogenic *E. coli* (ExPEC), particularly uropathogenic *E. coli* (UPEC) and neonatal meningitis *E. coli* (NMEC), raising concerns about potential zoonotic transmission from poultry to humans [11]. ExPEC, including APEC, can cause a range of extraintestinal infections in humans, such as urinary tract infections, neonatal meningitis, and sepsis [12], making them important to both public health and the poultry industry.

The global rise of infections caused by microorganisms resistant to antimicrobials, including first-line drugs such as cephalosporins, colistin, carbapenems, and fluoroquinolones, is a critical global problem, leading to serious morbidity, mortality, and health costs and a significant decrease in livestock production [13]. It is predicted that by 2050, antimicrobial resistance (AMR) will become the leading cause of death, according to global estimation, indicating that AMR caused over 1.2 million deaths in 2019, and it is projected to reach about 10 million annually by 2050 [14]. Furthermore, the resistant bacteria from poultry and other food animal sources have been on the increase around the world [15].

Antibiotic resistance is considered a key factor in the development of infection, as a virulence-like factor in specific ecological niches, where antibiotic-resistant bacteria are able to proliferate. This is particularly true in the hospital settings (intensive care units, burn units, etc.), where drug-resistant opportunistic pathogens can spread and cause disease more easily [16]. In Enterobacterales, especially in *E. coli* and *Klebsiella pneumoniae*, resistance to β-lactams due to production of extended-spectrum β-lactamases (ESBL) is a significant resistance mechanism that prevents the success of antimicrobial treatment of infections caused by these bacteria [17].

ESBL-producing bacteria have become a major worry in clinical settings worldwide, causing outbreaks related to enzymes of the CTX-M class being the most common, particularly CTX-M-15 [18]. Recently, among different Enterobacteriaceae members globally, ESBLs of the CTX-M type have supplanted SHV- and TEM-ESBLS types [19]. ESBL-producing isolates exhibited multidrug-resistant (MDR) effects to antimicrobials like aminoglycosides, tetracyclines, chloramphenicol, trimethoprim, sulfonamides, and quinolones [20].

Although carbapenems are considered the drug of choice for bacterial infections, resistance to carbapenems has now become a major clinical worry worldwide. This type of resistance is caused by enzymes hydrolyzing carbapenems, and the most influential carbapenemases on the hydrolysis of carbapenems and their geographic distribution are KPC, VIM, IMP, NDM, and OXA-48 types [21]. The presence of transferable genetic elements, such as plasmids, transposons, and integrons, plays a crucial role in the spread of multiple resistance genes [2].

Poultry is considered a highly significant source for the spread of antibiotic-resistant bacteria in the population and environment. Pathogenic *E. coli* in poultry is a direct menace to both poultry production and human health due to difficulty in treating infections [12].

The spread of antimicrobial-resistant bacteria has emerged as a One Health concern, as these bacteria can be transferred among humans, animals, and the environment. The use of antimicrobials in any of these sectors contributes to the increasing AMR burden [22]. Research indicates that AMR bacteria and their resistance determinants can be transmitted from food-producing animals to humans through direct contact or consumption of animal-derived products [23].

The objective of this study was to assess the prevalence of multidrug-resistant *E. coli* isolates from broilers intended for human consumption. It also aimed to characterize the genotype of antimicrobial resistance, phylogenetic groups, virulence factors, and integrons in these *E. coli* isolates.

## 2. Results

In our study, 111 *E. coli* isolates were obtained from MacConkey agar supplemented with cefotaxime (CTX 1 µg/mL) from the ceca of chicken broilers from three suppliers, distributed as 51 samples from supplier A, 30 from supplier B, and 30 from supplier C. The isolates were investigated for the determination and characterization of antibiotic resistance.

### 2.1. Antimicrobial Resistance Pattern

Antimicrobial susceptibility testing showed high resistance to amoxicillin (100%), cephalothin (96.4%), piperacillin (93.7%), cefotaxime (83.8%), cefepime (82%), tetracycline (79.3%), streptomycin (79.3%), aztreonam (77.5%), ceftazidime (75.7%), cefuroxime (73.9%), chloramphenicol (71.2%), and florfenicol (64.9%). Low resistance in *E. coli* isolates against cefotixin and colistin was observed, as the resistance average was lower than 20% (Figure 1). All isolates were multidrug-resistant (MDR). ESBL phenotype was confirmed in 72.07% (80/111) of isolates.

### 2.2. Detection of β-Lactamases Genes and Colistin Resistance Genes

The identification of resistance genes by PCR revealed that the most common broad-spectrum β-lactamase genes in the isolates were *bla*_CTX-M-G1_ (n = 85), followed by *bla*_TEM_ (n = 47) and *bla*_SHV_ (n = 36). The carbapenem resistance genes *bla*_OXA-48_ and *bla*_IMP_ were detected in fourteen (12.6%) and seven (6.3%) isolates, respectively. Eleven isolates were positive for the cephalosporinase CMY gene, and six isolates carried the plasmid-mediated colistin resistance *mcr-1* gene (Table 1).

Four β-lactamase genes (*bla*_CTX-M-G1_, *bla*_SHV_, *bla*_TEM_, *bla_CMY_*) *and* (*bla*_CTX-M-G1_, *bla*_SHV_, *bla*_TEM_, *bla_CMY_*) were detected in two strains. Three β-lactamase genes 3 (*bla*_CTX-M-G1_, *bla*_TEM_, *bla*_OXA-48_), 3 (*bla*_CTX-M-G1_, *bla*_TEM_, *bla*_IMP_), 2 (*bla*_CTX-M-G1_, *bla*_SHV_, bla_OXA-48_), 2 (*bla*_CTX-M-G1_, *bla*_SHV_, *bla*_TEM_), 1(*bla*_SHV_, *bla*_TEM_, *bla*_CMY_), 1 (*bla*_CTX-M-G1_, *bla*_TEM_, *bla*_CMY_), 1 (*bla*_SHV_, *bla_CMY_*, *bla*_OXA-48_), and 1 (*bla*_TEM_, *bla_CMY_*, *bla*_OXA-48_) were present in 14 *E. coli* isolates (Appendix A).

### 2.3. Identification of Antimicrobial Resistance Genes

A variety of non-β-lactam antimicrobial resistance genes were found in the isolates. The plasmid-mediated quinolone resistance genes *qnrS*, *qnrB*, and *qnrA* were found in twenty-four (21.6%), twenty-two (19.8%), and two (1.8%), respectively. The aminoglycosides (gentamycin and streptomycin) resistance genes *aac(3)-II*, *aac (6)-Ib-cr*, *aadA-1*, and *aadA-5* were found in 30 (27%), 32 (28.8%), 80 (72%), and 33 (29.7%), respectively. Twenty-two (19.8%) of the *E. coli* isolates harbored *sul* genes [*sul1* and *sul2*] (Appendix A, Table 1).

### 2.4. Virulence Factors Genes

A total of 17 virulence factors were identified in 111 *E. coli* isolates, with *traT*, *fimH*, and *iutA* having the highest detection rates as they appeared in two-thirds of the isolates. Among the adhesion factors, *fimH* was the most dominant (83.8%), followed by *bfpA* (43.2%), *eae* (11.7%), *papGIII* (7.2%), and *sfa/focDE* (3.6%). Toxin genes *stx-1*, *stx-2*, *cdt3*, and *ehxA* were detected in 38.7%, 7.2%, 2.7%, and 2.7%, respectively. Among the invasion factors, *traT* was the most predominant (83.8%), followed by *iutA* (73.9%) and *ibeA* (0.9%).

### 2.5. Phylogenetic Groups

The phylogenetic analysis classified the *E. coli* isolates into A 43 (38.7%), B1 18 (16.2%), F 16 (14.4%), D 15 (13.5%), E 8 (7.2%), B2 6 (5.4%), C 3 (2.7%), and clade I 2 (1.8%).

### 2.6. Association of Virulence Genes and Resistance Genes Distribution Among Phylogenetic Groups

The analysis of phylogroups distribution among the resistance genes revealed that the resistance genes *bla*_CTX-M-1_, *bla*_TEM_, *bla*_SHV_, *bla*_OXA-48_, *qnrB*, and a*ac(6)Ib-cr* were distributed among phylogenetic groups, whereas *mcr-1*, *bla*_IMP_, *qnrS*, and *qnrA* were present only in groups A and B1. β-lactamase genes CTX-M-ESBL, TEM, and SHV were the most dominant in the A group, followed by the F and B1 groups, with a statistically significant difference (*p* value = 0.003). In addition, quinolone resistance genes *qnrS* and a*ac(6)Ib-cr* were classified in the A1 and D groups (*p* value = 0.003 and 0.054, respectively) (Table 2 and Figure 2).

Figure 3 shows the association between virulence genes and phylogenetic groups. The genes *traT*, *fimH*, *BfpA*, *iutA*, *fyuA*, *papGIII*, *stx-1*, *hly*, and *cnf1* were widely disseminated in the groups. Genes *stx-2*, *eae*, and *cdt3* were found in groups A, B1, and B2. Group A was the dominant group that carried the virulence genes, whereas *ehxA* and *sfa/focDE* were less present in group A isolates.

### 2.7. Characterization of Integrons

Class 1 integrons were found in 92 (82.9%) of the isolates, and the 3′CS that contains *qacEΔ1* and *sul1* was detected in 43 (38.7%) of positive class 1 integrons. Class 2 integrons were demonstrated in 44 (39.6%) isolates. Thirty-four isolates harbored both integrons. The sequencing of variable regions (VR) of class 1 and 2 integrons revealed different gene cassette arrangements. In VR of class 1 integrons, gene cassettes containing genes encoding resistance to trimethoprim (*dfrA*) and streptomycin (*aadA*) were detected in three different arrangements: *dfrA1-aadA1* (eight isolates), *dfrA12-aadA2* (five isolates), and *aadA1* (three isolates). For the VR of class 2 integrons, the gene cassettes *sat2-dfrA1* and *sat2-aadA1* were detected in nine and three isolates, respectively (Appendix A).

## 3. Discussion

Poultry and poultry products are mostly popular and consumed foods by humans worldwide. However, few data are available regarding the molecular characterization of *E. coli* isolated from poultry in Tunisia. In this study, we assessed the molecular characterization of *E. coli* isolates collected from the ceca of chicken broilers in poultry slaughterhouses from three main geographical regions, extending from Great Tunis in the north, Mahdia in the central–east, and Sfax in the southeast of Tunis.

Our findings revealed that the broilers intended for human consumption could be contaminated with broad-spectrum antibiotic-resistant *E. coli*, and the possible spread of these strains to humans and the environment is alarming. This is a big threat for Tunisia as a developing country that has limited sufficiently qualified staff and challenges of the economic situation that impact health situations [24].

The ESBL rate among *E. coli* was confirmed in 80 isolates (72.07%). This rate is consistent with the frequency of ESBL production (76.5%) reported in *E. coli* isolates from chicken meat in Tunisia [25]. This finding was similar to that of Liu et al. [26] and higher than the rates of ESBL-producing *E. coli* detected in chicken from Ghana, Algeria, and Tunisia, which were 56.2%, 55.5%, and 20.1%, respectively [27,28,29], as well as in Egypt, where the frequency related to chicken meat was 61.6% [17]. ESBL production has also been reported in cloacal swabs of chickens in recent studies from Malaysia (37.2%) [30] and Taiwan (36.3%) [31]. The comparatively higher rate observed in our isolates may be attributed to the fact that they were collected from the cecum, which generally yields a high frequency of bacterial isolates and resistance traits within the intestinal environment. A Korean study found an occurrence rate of ESBL in chicken of about 94.1% [32]. Many studies confirmed that the high prevalence of ESBL-producing *E. coli* in broilers is becoming common and more significant than in other food animals. They also revealed that the chicken origin of ESBL–*E. coli* is related to more resistance to antibiotics compared with ESBL from other food-producing animals [26,33]. From the previous findings, the ESBL–*E. coli* rates may differ by the sample sources and region. The high ESBL rate in *E. coli* from poultry is driven by excessive antibiotic use, facilitating resistance selection. Horizontal gene transfer via plasmids accelerates spread, while poor biosecurity, contaminated feed, and environmental persistence further sustain transmission [30,34].

This study reported a high percentage of resistance compared with another Tunisian study conducted in *E. coli* isolated from 150 healthy poultry, with resistance rates to tetracycline, streptomycin, sulfamethoxazole/trimethoprim, nalidixic acid, ciprofloxacin, amoxicillin/clavulanic acid, cefotaxime, and ceftazidime of about 74.7%, 57%, 57%, 54.4%, 34.2%, 15.2%, 5.1%, and 3.8%, respectively [35]. The higher resistance observed in our study may indicate an increased use of antimicrobials in veterinary medicine in recent years, potentially driving the selection and spread of resistant *E. coli* strains. In Tunisia, a broad range of antimicrobials is used in food-producing animals, including poultry, for both prophylactic and therapeutic purposes. Commonly administered antibiotics include penicillins (e.g., amoxicillin), tetracyclines, macrolides, colistin, sulfonamides, aminoglycosides, and quinolones, raising concerns about antimicrobial resistance (AMR) and its potential transmission to humans via the food chain and environment.

A study conducted in Qatar recorded high resistance in *E. coli* isolated from broiler chickens; the percentage of resistance to antibiotic agents (ampicillin, cephalothin, ciprofloxacin, tetracycline, and fosfomycin) in strains was high (76.5–100%), and MDR isolates were 99.3% [11]. In another study from South Korea, they observed highly resistant *E. coli* isolates to cefotaxime, cefazolin, ampicillin, and piperacillin from fecal samples of chickens collected from the intestinal tracts at slaughterhouses, and all isolates were MDR [32].

All of the isolates were MDR, which indicates that the *E. coli* isolates may be a reservoir of antibiotic resistance. The use of antimicrobials for disease treatment and prophylaxis in poultry farming may lead to an increase in bacterial resistance. Hundreds of tons of antibiotics are used annually in the agricultural industry [36]. The high frequency of MDR strains is considered a significant health risk because these isolates may disseminate to food products and the environment and, thus, be transferred to humans. Colistin is a last-resort antibiotic in the treatment of MDR Gram-negative bacteria, and fortunately, colistin was susceptible against 92.8% of our isolates. Consequently, this antibiotic may be suggested as a drug of choice for MDR *E. coli* in our country.

The CTX-M family encoding extended-spectrum β-lactamases is dominant and has become a major worry in health settings worldwide [37]. In this study, the *bla*_CTX-M_ gene was found in 86 (77.5%) strains. Similar findings were reported in Ghana, where the *bla*_CTX-M_ gene was the most dominant and confirmed in 95.1% of the strains [27]. The *bla*_TEM_ and *bla*_SHV_ genes encoding β-lactamase have been reported in 47 (42.3%) and 36 (32.4%) strains, respectively. The predominance of the CTX-M gene over other β-lactamases genes was recorded in a previous Tunisian study from healthy poultry [38]. However, many studies reported the dominance of the *bla*_TEM_ and *bla*_SHV_ genes over *bla*_CTX-M_ among *E. coli* from chicken in Algeria, Egypt, and Poland [39,40,41].

The plasmid-mediated colistin resistance determinant (*mcr-1*) is associated with colistin resistance and is widely detected in different bacteria isolated from humans, food, and livestock [42]. In our study, *mcr-1* was detected in six colistin-resistant strains with a prevalence rate of 5.4% (six of all tested isolates). This finding is lower than in previous reports from Malaysia and Egypt [40,43], which found *mcr-1* determinant in 27.6%, 41.83%, and 64.3% of the tested *E. coli* strains, respectively, and they suggested that chicken farms may be a source of colistin-resistant *E. coli*. A Tunisian study reported *mcr-1* from chicken gut microbiota in Tunisia [42]. It has been demonstrated that most *mcr-1*-positive bacteria include various lactamase genes, such as *bla*_CTX-M_, *bl*a_OXA-48_, *bl*a_TEM_, *bl*a_SHV,_ and *bl*a_CMY_. This combination further restricts the therapeutic alternatives because the last two last-resort medications, colistin and carbapenems, are challenged by this occurrence.

The current study reported the carbapenem resistance genes *bla*_OXA-48_ and *bla*_IMP_ in fourteen (12.6%) and seven (6.3%) isolates, respectively. These genes were confirmed with other β-lactamases, particularly CTX-M in the same isolate, and, in some cases, with TEM, SHV, and CMY. There are a few reports about the prevalence of carbapenem resistance in food-producing animals in Tunisia. A recent study reported *bla*_OXA-48_ and *bla*_IMP_ genes among ESBL-producing *E. coli* isolates from Tunisian diarrheic calves in Tunisia [44]. Another Tunisian study documented the presence of *bla*_VIM_ and *bla*_IMP_ genes in *E. coli* isolates from healthy rabbits [45]. In an Egyptian study, Hamza et al. [46] investigated *bla*_NDM_, *bla*_KPC_, and *bla*_OXA48_ genes among broiler chickens and reported 15% of *Klebsiella pneumoniae* resistant to carbapenems, with all the isolates harboring the *bla*_NDM_ gene. The presence of carbapenem resistance genes *bla*_NDM_, *bla*_OXA_, and *bla*_IMP_ was reported in *E. coli* from chickens in poultry farms in Malaysia [47]. While carbapenems are not administered in poultry or other food animal production, resistance to them in *E. coli* isolates could be linked to resistance against other antibiotics commonly used there. These co-resistant bacteria might be disseminated through direct contact, insect vectors, or other animals [48,49].

The simultaneous production of four or three βlactamase types in the same strain, consisting of ESBLs (CTX-M, SHV, TEM), cephalosporinase (CMY), and carbapenemases (OXA-48 and IMP), is remarkable in 16 (14.4%) *E. coli* isolates. The existence of more than two βlactamase genes, ESBLs, and carbapenemases in the same isolate has been reported in Enterobacteriaceae from several sources in Tunisia [50,51,52]. The coexistence of multiple β-lactamase genes within the same bacterium is concerning, as it indicates the presence of important resistance determinants and may predict the emergence and dissemination of multidrug-resistant (MDR) strains. This could ultimately lead to treatment failures, resulting in significant morbidity and mortality.

We screened genes encoding antimicrobial resistance in the isolates that exhibited phenotype resistance in the disc diffusion test. Increased use of quinolones is associated with the development of quinolone resistance issues in humans and veterinary medicine [53]. Plasmid-mediated quinolone resistance (PMQR) genes, which cause quinolone resistance, were detected in 44% (49/111), with *aac (6)-Ib-cr* being the most frequent gene, followed by *qnrS* and *qnrB.* The presence of PMQR genes is linked with MDR plasmids, mostly associated with β-lactamases genes [54]. Furthermore, these genes increase resistance to other antimicrobials like aminoglycosides, β-lactams, chloramphenicol, sulfonamides, tetracyclines, and trimethoprim [41]. The sulfonamide resistance genes *sul1* and *sul2* were detected in 20 (18%) and 21 (18.9%), respectively. In agreement with our study, these elements were widely distributed in *E. coli* from poultry chicken [41,55]. These genes were more prevalent in isolates containing integrons than integron-negative isolates, with a statistically significant difference. This finding is consistent with reports from Tunisia and Algeria [39,55].

One of the significant findings of this study is the virulence factors (VFs). Several VFs associated with extraintestinal pathogenic *E. coli* (ExPEC) were identified in the isolates. All isolates had one or more genes. These factors are common to ExPEC strains in animals and humans, allowing these bacteria to colonize, invade, and lyse the host cell with toxins and cause infections outside the gut [56]. The most frequent genes were *traT*, *fimH*, and *iutA*, followed by *ibeA*, *papGIII*, *cdt3*, and *sfa/focDE.* Other studies reported similar VFs in *E. coli* isolated from poultry. The *fimH* and *iutA* were detected in Nigeria and Zimbabwe [57,58], *iutA*, *papC*, and *sfa/focDE* in Brazil [56], and *iutA* and *ibeA* in France [59]. The VFs associated with diarrheagenic *E. coli* (*stx-1*, *stx-2*, *eae*, and *ehxA*) were confirmed among the isolates, similar to findings in a previously reported study [8]. These factors enhance the capacity of *E. coli* to cause severe diseases in humans due to their ability to excrete toxins [60]. Our isolates contained the genetic content of the ExPEC, and this increases the emerging challenges to public health because they can be transmitted to humans by contaminated meat, knowing that bacteria were isolated from healthy chicken intended for slaughter. These strains belong to gut microbiota, whereas ExPEC has the ability to invade organs and cause infections outside the intestine in humans and animals by possessing these virulence factors [56]. The gene *iutA* encoding a protein associated with increased serum survival in humans and animals and colonizing capacity was found in 73.9% of our isolates. The *TraT* gene expresses a lipoprotein located in the outer membrane that inhibits the classical pathway of complement activation and leads to serum survival [3], which was found with a high frequency in 85.6% of the strains. This gene was also the most detected in *E. coli* isolated from hens at three industrial laying hen farms in the north of Tunisia [61]. Studies indicated that poultry may be a source of ExPEC for humans, in the case of consumption of highly contaminated meat. The similarity of genetic relatedness between isolates from healthy poultry and processed chicken products suggests the spread of these bacteria via the food chain [62].

Class 1 integrons were identified in 92 (82.9%) isolates, and class 2 integrons in 44 (39.6%), and our finding is consistent with other studies that indicated domination of class 1 among *E. coli* strains [61,63]. Integrons are a genetic system that contains a site-specific recombination, allowing integration, expression, and exchange of several gene cassettes that play a major role in the global dissemination of antibiotic resistance, particularly in *Enterobacteriaceae* [64]. Integrons can spread resistance genes from one organism to another, which poses a significant problem for the livestock and poultry sectors, given how poorly infection control systems operate [65]. We sequenced the VR in classes 1 and 2 integrons. In class 1, the VR revealed the presence of resistance genes to trimethoprim (*dfrA1&12*) and streptomycin (*aadA1&2*), and the gene cassette array *dfrA1-aadA1* was the most dominantly found. These observations are in agreement with the findings of a poultry study in Tunisia [55]. Gene cassettes in class 2 included resistance genes *dfrA1* and *aadA1* with the *sat2* gene. These determinants were detected in class 2 integrons worldwide [55,66]. Class 1 integrons are distinguished by a conserved segment at the 5’ end (known as 5CS), which contains the integrase gene (*intl1*), and a conserved segment at the 3’ end (known as 3CS), which includes the *qacEΔ* gene responsible for conferring resistance to quaternary ammonium compounds. Additionally, the 3CS also contains the *sul1* gene, which encodes resistance to sulfonamides [67]. This structure was found in 43 (38.7%) among class 1 positive isolates, and the results of the presence and lack of this structure were interesting; in fact, the deficient integrons missing some of these genes in the 3′CS region have been described [63,68].

One of the significant findings of this study is the detection of four β-lactamase genes (*bla*_CTX-M-G1_, *bla*_SHV_, *bla*_TEM_, *bla_CMY_*) *and* (*bla*_CTX-M-G1_, *bla*_SHV_, *bla*_TEM_, *bla_CMY_*) in two strains, along with the presence of class 1 and class 2 integrons carrying gene cassettes. This is a public health concern, as these bacteria also harbor other important resistance genes, such as *mcr-1*, in addition to multiple virulence genes, highlighting the alarming potential for their dissemination.

Clermont et al. improved the PCR method for phylogenetic typing, which has become widely used, allowing the differentiation of virulent strains (B2 or D) and commensal lineages (A and B1) [7]. The 111 CTX-resistant *E. coli* isolates were classified mainly in group A (38.7%) and in group B1 (16.2%), which were considered as commensal lineages. Similar results were found in Brazil [69], Portugal [70], and India [12]. In our results, 13.5% of the CTX-resistant isolates were classified in group D, 7.2% in group E, and 5.9% of isolates were assigned to the B2 category, which has the highest pathogenicity. These results agree with other studies [12,39]. The group B2 and D isolates are deemed ExPEC [8], while phylogroup E is linked to both human and animal intestinal strains, as well as human *E. coli* O157:H7 strains [71]. These findings propose that our strains belonging to the B2 and D groups are ExPEC, as well as the prospect of phylogroup E, which may be linked with ExPEC.

The limitations of this study include the absence of plasmid profiling to assess horizontal gene transfer and the lack of Multilocus Sequence Typing (MLST) to characterize bacterial isolates by determining their sequence types (STs). Accordingly, future investigations should incorporate these methods, particularly MLST, and analyze selected isolates using whole-genome sequencing for comprehensive characterization.

## 4. Material and Methods

### 4.1. Samples

Samples were collected from healthy broiler chickens intended for slaughter for human consumption. A total of 111 cecal samples were obtained during the first four months of 2022 from the three main poultry meat suppliers in Tunisia, distributed as follows: 51 samples from the first supplier, 30 from the second, and 30 from the third. These suppliers are the major providers of poultry meat, located in Greater Tunis, Sousse (central–east Tunisia), and Sfax (southeast Tunisia). The samples were collected randomly from healthy broiler chickens at approximately 35 days of age. They were kept at 4 °C and rapidly transported to the laboratory for bacterial analysis.

### 4.2. Bacterial Isolation and Identification

A gram of cecal contents from each sample was suspended in a 1.5 mL Eppendorf tube containing buffered peptone water and was incubated overnight at 37 °C. A 10 µL loop of bacterial suspension was plated onto MacConkey agar (MAC) (Oxoid, Basingstoke, UK). MAC was supplemented with cefotaxime (CTX 1 µg/mL) and incubated at 37 °C overnight, as previously described [72]. One bacterial colony was picked up from MAC supplemented with CTX culture and plated on brain heart infusion agar (Merck, Darmstadt, Germany) for further investigation, whereas the *E. coli* isolates from non-supplemented MAC plates were included for prevalence rate purposes. The bacterial isolates were identified by API 20E gallery (bioMérieux, Marcy-l’Étoile, France) and were confirmed by a PCR targeting the *uidA* gene (486 bp) using specific primers (Forward: 5′-ATCACCGTGGTGACGCATGTCGC-3′ and Reverse: 5′-CACCACGATGCCATGTTCATCTGC-3′) (Bio Basic, Markham, ON, Canada) based on the method previously described by [73]. Bacterial isolates were stored at −20 °C in brain heart infusion broth supplemented with 20% glycerol.

### 4.3. Antimicrobial Sensitivity Test and ESBL Phenotype Detection

Antimicrobial sensitivity was carried out by the disk diffusion method on Mueller-Hinton agar (BioRad, Marne la Coquette, France) plates according to the guidelines of the Antibiogram Committee of the French Society (CA-SFM) [74] using twenty antibiotic discs (Bio-Rad France) comprising μg/disk: twelve β-lactams represented by amoxicillin (25), amoxicillin/clavulanic acid (20/10), ticarcillin/clavulanic acid (75/10), cefotaxime (30), ceftazidime (30), cefepime (30), cefoxitin (30), aztreonam (30), ertapenem (10), piperacillin (30), cephalothin (30), cefuroxime (30), and nine non-β-lactams antibiotics represented by (chloramphenicol (30), gentamicin (15), nalidixic acid (30), enrofloxacin (5), tetracycline (30), sulfamethoxazole/trimethoprim (1.25/23.75), streptomycin (10), and florfenicol (30). Furthermore, colistin susceptibility was examined by Colispot [75].

The ESBL phenotype was detected by a synergy test using a disk of cefotaxime, ceftazidime, and cefepime around a disk containing clavulanic acid, which is placed on Mueller-Hinton agar plates with a distance of 30 mm between the two discs. The enhanced inhibition zone of any discs combined with clavulanic acid by 5 mm or more was considered an ESBL producer [76]. All isolates exhibited resistance to third-generation cephalosporins (cefotaxime and/or ceftazidime), and positive ESBL confirmation disk tests were selected for molecular analyses. Furthermore, the multidrug-resistant (MDR) percentage was searched. The MDR is known as an isolate exhibiting resistance to at least one agent in three or more antimicrobial classes [55]. The *E. coli* ATCC 25922 reference strain was used as a control.

### 4.4. Detection of β-Lactamase Genes

PCR amplification was performed to search for the presence of the β-lactamases encoding genes: *bla*_CTX-M,_
*bla*_TEM,_
*bla*_OXA,_ *bla*_SHV,_ and *bla*_CMY_. Carbapenem resistance genes *bla*_VIM_, *bla*_IMP_, *bla*_NDM_, *bla*_KPC,_ and *bla*_OXA-48_ were searched by PCR [21]. The primers used are listed in Appendix A.

### 4.5. Detection of Antimicrobial Resistance Genes to Non-β-Lactam Agents

The presence of genes that confer resistance to sulfamethoxazole (*sul1*, *sul2*, and *sul3*), aminoglycosides (*aac(3)-I*, *aac(3)-II*, *aac(3)-IV*, *AadA-1* and *AadA-5*), quinolones (*qnrA*, *qnrB*, *qnrS*, and *aac*(6’)-1b, and colistin (*mcr-1*)) were tested by PCR amplification [77].

### 4.6. Detection and Characterization of Integrons

The presence of *intI1* and *intI2* genes was screened in all isolates by PCR. The presence of *qacED1*-*sul1* genes within 3’-conserved regions of class 1 integrons was identified by PCR. The gene cassettes in the variable region of class 1 integrons were screened by PCR and sequencing. The primers used are listed in Appendix A [77].

### 4.7. Determination of Phylogenetic Groups and Virulence Factors

Using the revised technique previously described by Clermont, Christenson, Denamur, and Gordon [7] and the primers given in Appendix A, the phylogenetic groups (A, B1, B2, D, C, E, and Clade 1) to which the strains belong were identified.

The presence of virulence genes *stx1*, *stx2*, *ehxA*, *eae*, *cdt3*, *cnf1*, *hly*, *aer*, *papA*, *bfpA*, *papG allele III*, *fimH*, *traT*, *ibeA*, *sfa/foc*, *iutA*, and *fyuA* was searched by PCR (Appendix A).

Bacterial strains carrying the target genes, previously isolated from animals in the Laboratory of Microbiology and Immunology at the National School of Veterinary Medicine, Tunisia, were used in this study as positive controls for antimicrobial resistance genes or virulence genes.

### 4.8. Data Analysis

Statistical Package for the Social Sciences (SPSS) was used to analyze the data and determine the frequency of bacterial isolates, resistance genes, virulence factors, and phylogenetic groups. Pearson’s Chi-square test was used to investigate the significant differences in phylogroups distribution among the resistance genes and virulence genes, using SPSS version 26 software (IBM Corporation, Somers, NY, USA). The level of statistical significance was set at *p* < 0.05.

## 5. Conclusions

The high rate of antimicrobial resistance and the development of MDR *E. coli* in poultry intended for human consumption, as well as the spreading risk of these bacteria to humans and the environment, are concerning. This study proved a high prevalence of ESBL-producing *E. coli* in chicken. It also demonstrated a high level of *bla*_CTX-M_-group1-producing isolates. Chicken farms may be a significant source of the ESBL-producing bacteria, causing illnesses in people that are difficult to cure. The presence of multiple antimicrobial resistance genes and several virulence factors associated with integrons makes *E. coli* a potential food safety issue and poses a significant threat to public health. This is due to the possibility of horizontal transfer of these genes to bacteria in the environment and humans, complicating the concern over antibiotic resistance. Limiting the use of antimicrobials in chicken production may help minimize the emergence of antimicrobial resistance genes and virulence factors, hence lowering the risk of human infection. Regulatory control of the administration of antimicrobials is required to prevent the development of antibiotic-resistant strains and to safeguard public health.

## Figures and Tables

**Figure 1 antibiotics-14-00931-f001:**
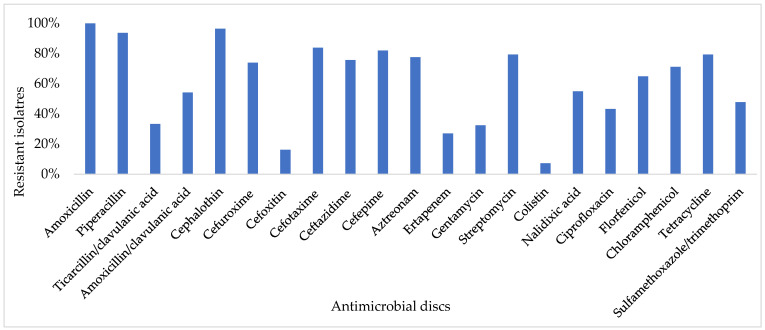
Antimicrobial resistance of 111 *E. coli* isolates.

**Figure 2 antibiotics-14-00931-f002:**
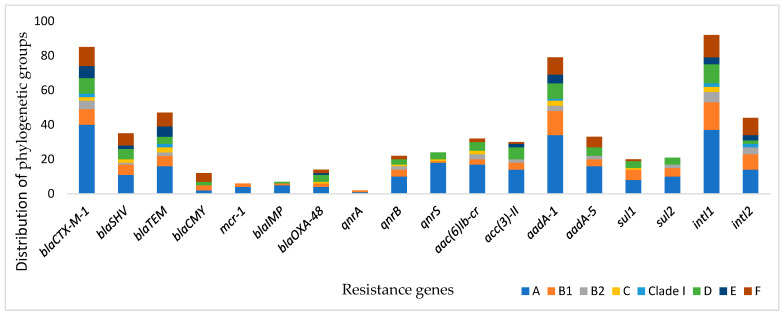
Prevalence of resistance genes and their distribution according to phylogenetic groups.

**Figure 3 antibiotics-14-00931-f003:**
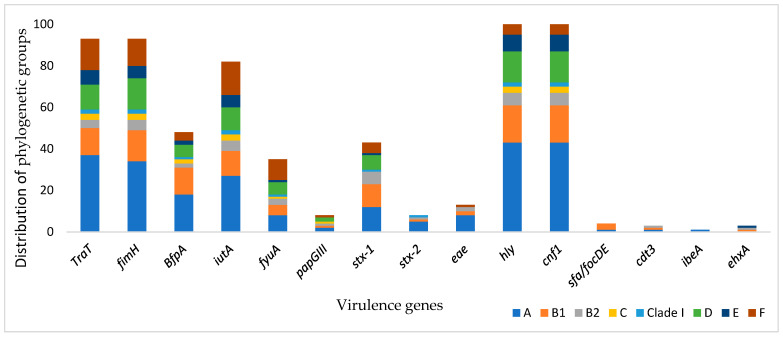
Prevalence of the virulence genes and distribution of phylogenetic groups.

**Table 1 antibiotics-14-00931-t001:** Frequency of the detected genes in 111 *E. coli* isolates obtained from the ceca of chicken broiler samples.

Category	Determinant	n (%)
Virulence genes	*traT*	93 (83.8%)
	*fimH*	88 (79.3%)
	*iutA*	82 (73.9%)
	*bfpA*	48 (43.2%)
	*fyuA*	35 (31.5%)
	*stx-1*/shiga toxin	34 (30.6%)
	stx-2	8 (7.2%)
Resistance genes	*bla* _CTX-M-G1_	85 (76.6%)
	*bla* _TEM_	47 (42.3%)
	*bla* _SHV_	36 (32.4%)
	*bla* _CMY_	12 (10.8%)
	*bla* _IMP_	7 (6.3%)
	*bla* _OXA-48_	14 (12.6%)
	*mcr-1*	6 (5.4%)
	*aadA-1*	80 (72%)
	*aadA-5*	33 (29.7%)
	*aac (6)-Ib-cr*	32 (28.8%)
	*qnrS*	24 (21.6%)
	*qnrB*	22 (19.8%)
Integrons	*Int1+*	92 (82.9%)
	*Int2+*	44 (39.6%)

**Table 2 antibiotics-14-00931-t002:** Prevalence of resistance genes and their distribution according to phylogenetic groups.

Genes	A (n = 43)	Percent (%)	B1 (n = 18)	Percent (%)	B2 (n = 6)	Percent (%)	C (n = 3)	Percent (%)	Clade I (n = 2)	Percent (%)	D (n = 15)	Percent (%)	E (n = 8)	Percent (%)	F (n = 16)	Percent (%)	Total	Percent (%)	*p*-Value
*bla* _CTX-M-1_	40	93	9	50	5	83.3	2	66.7	2	100	9	60	7	87.5	11	68.8	85	76.6	0.001
*bla* _SHV_	11	25.6	6	33.3	1	16.7	2	66.7	0	0	6	40	2	25	7	43.8	36	32.4	0.001
*bla* _TEM_	16	37.2	6	33.3	2	33.3	3	100	2	100	4	26.7	6	75	8	50	47	42.3	0.001
*bla* _CMY_	2	4.7	3	16.7	0	0	0	0	0	0	2	13.3	0	0	5	31.3	12	10.8	0.156
*mcr-1*	4	9.3	2	11.1	0	0	0	0	0	0	0	0	0	0	0	0	6	5.4	0.741
*bla* _IMP_	5	11.6	1	5.6	0	0	0	0	0	0	1	6.7	0	0	0	0	7	6.3	0.819
*bla* _OXA-48_	4	9.3	2	11.1	0	0	1	33.3	0	0	4	26.7	1	12.5	2	12.5	14	12.6	0.661
*qnrA*	1	2.3	1	5.6	0	0	0	0	0	0	0	0	0	0	0	0	2	1.8	0.975
*qnrB*	10	23.3	4	26.7	2	33.3	1	33.3	0	0	3	20	0	0	2	12.5	22	19.8	0.705
*qnrS*	18	41.9	1	5.6	0	0	1	33.3	0	0	4	26.7	0	0	0	00	24	21.6	0.003
*aac(6)Ib-cr*	17	39.5	3	16.7	3	50	2	66.7	0	0	5	33.3	0	0	2	12.5	32	28.8	0.054
*acc(3)-II*	14	32.6	4	26.7	2	33.3	0	0	0	0	7	46.7	2	25	1	6.3	30	27	0.011
*aadA-1*	34	79.1	14	77.8	3	50	3	100	1	50	9	60	5	62.5	10	62.5	80	72	0.001
*aadA-5*	16	37.2	4	26.7	2	33.3	0	0	0	0	5	33.3	0	0	6	37.5	33	29.7	0.001
*sul1*	8	18.6	6	33.3	0	0	1	33.3	0	0	4	26.7	0	0	1	6.3	20	18	0.27
*sul2*	10	23.3	5	27.8	2	33.3	0	0	0	0	4	26.7	0	0	0	0	22	19.8	0.231
*IntI1*	37	86	16	88.9	6	100	3	100	2	100	11	73.3	4	50	13	81.3	92	82.9	0.001
*IntI2*	14	32.6	9	50	4	66.7	0	0	2	100	2	13.3	3	37.5	10	62.5	44	39.6	0.001

## Data Availability

The datasets generated and analyzed during the current study are included within this manuscript.

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
