# Peer review of "Phenotypic and Molecular Study of Multidrug-Resistant Escherichia coli Isolates Expressing Diverse Resistance and Virulence Genes from Broilers in Tunisia"

_antibiotics, 2025, doi:10.3390/antibiotics14090931_

Round 1
Reviewer 1 Report
Comments and Suggestions for Authors
This is a comprehensive study addressing the phenotypic and molecular characteristics of multidrug-resistant (MDR) E. coli in poultry in Tunisia, with clear relevance to public health and One Health perspectives. The research methods and scope are strong, and the findings are significant. However, the abstract and some sections would benefit from structural and language clarity improvements to better highlight key results and significance.
Comments and Suggestions
-
Some sentences are too long and should be split for clarity. Change “This study aimed to investigate and characterize at the molecular and phenotypic levels the prevalence of antimicrobial resistance in Escherichia coli isolates from caeca of healthy broilers.” to “This study aimed to investigate the prevalence of antimicrobial resistance in Escherichia coli isolated from the caeca of healthy broilers. Both molecular and phenotypic characterizations were conducted.”
-
Replace "several E. coli was isolated..." to “Several E. coli strains were isolated...”
-
Change "The most fequently" to "most frequently"
-
Change “Class 2 integron was demonstrated in 39.6% isolates.” →to “Class 2 integrons were detected in 39.6% of isolates.”
-
For more consistency, please use consistent terminology: e.g., "β-lactamase gene" vs. "β lactamase gene"
-
Use percentages in a readable format: e.g., "83.8%" instead of repeating long lists.
-
The abstract is dense and data-heavy. It might be better to focus only on key findings and move technical detail (e.g., exact percentages of each gene) to the results section. So rewritten for Clarity and Flow as follows;
"This study investigated the molecular and phenotypic characteristics of antimicrobial resistance in Escherichia coli isolates recovered from the caeca of healthy broilers in Tunisia. A total of 111 E. coli isolates were obtained from chicken samples collected at slaughterhouses and cultured on cefotaxime-supplemented MacConkey agar. All isolates exhibited a multidrug-resistant (MDR) phenotype, and 72.1% were confirmed as extended-spectrum β-lactamase (ESBL) producers. The most frequent β-lactamase gene was blaCTX-M-G1, followed by blaTEM and blaSHV. Carbapenem resistance genes (blaOXA-48 and blaIMP) were detected in 12.6% and 6.3% of isolates, respectively, while six isolates harbored the colistin resistance gene mcr-1.
Among the tested virulence genes, fimH, traT, and iutA were the most prevalent, detected in over 70% of isolates. Class 1 integrons were present in 83% of isolates, and class 2 integrons in 39.6%, with gene cassettes encoding resistance to trimethoprim (dfrA) and streptomycin (aadA).
These findings highlight the widespread presence of MDR and ESBL-producing E. coli strains with virulence traits and integrons in poultry, underscoring the risk of transmission to humans. This study provides essential data supporting the implementation of integrated surveillance strategies in line with the One Health approach."
8. The suggestion for Methodology
-
Consider providing information on clonal relationships (e.g., MLST) to better assess dissemination patterns.
-
It would strengthen the impact to compare findings with antibiotic usage data in poultry in Tunisia if available.
-
Consider discussing biosecurity practices in Tunisian poultry farms and their possible link to resistance spread.
The English could be improved to more clearly express the research.
Author Response
Thank you for your valuable comments of our manuscript entitled “Phenotypic and Molecular Study of Multidrug-Resistant Escherichia coli Isolates Expressing Diverse Resistance and Virulence Genes from Broilers in Tunisia” Manuscript Number.: antibiotics-3799029. Please find attached our responses to the comments from the Reviewers. We have incorporated the necessary changes in the manuscript, which are highlighted in yellow.
Best regards,
Dr. Ghassan Tayh
Point #1: Some sentences are too long and should be split for clarity. Change “This study aimed to investigate and characterize at the molecular and phenotypic levels the prevalence of antimicrobial resistance in Escherichia coli isolates from caeca of healthy broilers.” to “This study aimed to investigate the prevalence of antimicrobial resistance in Escherichia coli isolated from the caeca of healthy broilers. Both molecular and phenotypic characterizations were conducted.”
Response: As suggested by the reviewer the correction has been made in the revised manuscript: (Please see Page No. 1 in lines 18-20).
Point #2: Replace "several E. coli was isolated..." to “Several E. coli strains were isolated...”
Response: As suggested by the reviewer the correction has been made in the revised manuscript: (Please see Page No. 1 in lines 20).
Point #3: Change "The most fequently" to "most frequently"
Response: As suggested by the reviewer the correction has been made in the revised manuscript: (Please see Page No. 1 in lines 26-27).
Point #4: Change “Class 2 integron was demonstrated in 39.6% isolates.” →to “Class 2 integrons were detected in 39.6% of isolates.”
Response: As suggested by the reviewer the correction has been made in the revised manuscript: (Please see Page No. 1 in line 29).
Point #5: For more consistency, please use consistent terminology: e.g., "β-lactamase gene" vs. "β lactamase gene"
Response: As suggested by the reviewer the correction has been made in the revised manuscript.
Point #6: Use percentages in a readable format: e.g., "83.8%" instead of repeating long lists.
Response: Done.
Point #7: The abstract is dense and data-heavy. It might be better to focus only on key findings and move technical detail (e.g., exact percentages of each gene) to the results section. So rewritten for Clarity and Flow as follows;
"This study investigated the molecular and phenotypic characteristics of antimicrobial resistance in Escherichia coli isolates recovered from the caeca of healthy broilers in Tunisia. A total of 111 E. coli isolates were obtained from chicken samples collected at slaughterhouses and cultured on cefotaxime-supplemented MacConkey agar. All isolates exhibited a multidrug-resistant (MDR) phenotype, and 72.1% were confirmed as extended-spectrum β-lactamase (ESBL) producers. The most frequent β-lactamase gene was blaCTX-M-G1, followed by blaTEM and blaSHV. Carbapenem resistance genes (blaOXA-48 and blaIMP) were detected in 12.6% and 6.3% of isolates, respectively, while six isolates harbored the colistin resistance gene mcr-1.
Among the tested virulence genes, fimH, traT, and iutA were the most prevalent, detected in over 70% of isolates. Class 1 integrons were present in 83% of isolates, and class 2 integrons in 39.6%, with gene cassettes encoding resistance to trimethoprim (dfrA) and streptomycin (aadA).
These findings highlight the widespread presence of MDR and ESBL-producing E. coli strains with virulence traits and integrons in poultry, underscoring the risk of transmission to humans. This study provides essential data supporting the implementation of integrated surveillance strategies in line with the One Health approach.
Response: We thank the reviewer for the comments and we changed the abstract as suggested by the reviewer in the revised manuscript.
Point #8: The suggestion for Methodology
Consider providing information on clonal relationships (e.g., MLST) to better assess dissemination patterns. It would strengthen the impact to compare findings with antibiotic usage data in poultry in Tunisia if available. Consider discussing biosecurity practices in Tunisian poultry farms and their possible link to resistance spread.
Response: We agree with the reviewer’s suggestion; however, MLST was not performed in this study. We acknowledge this as a limitation and have addressed it in the manuscript.
Regarding antibiotic use in the poultry sector in Tunisia, we have added a paragraph on this topic to the revised manuscript (see Page 9, lines 242–247).
As for biosecurity practices in Tunisian poultry farms, we recognize their importance in understanding the dissemination of bacterial strains and antimicrobial resistance. However, this aspect was not discussed in our study since our focus was on isolates collected from slaughterhouses, specifically from chickens intended for human consumption.

Reviewer 2 Report
Comments and Suggestions for Authors
General Overview:
The authors present a study focusing on the phenotypic and molecular characterization of multidrug-resistant (MDR) Escherichia coli isolated from broiler chickens in Tunisia. The aim is relevant and timely, particularly within the context of global concerns regarding antimicrobial resistance in food-producing animals. However, several methodological and presentation-related issues require clarification and improvement.
Major Comments
1. Materials and Methods (Lines 130–136):
The methodology describes the isolation of E. coli from caecal content using MacConkey agar (MAC) and MAC supplemented with cefotaxime (1 μg/ml). A single colony was selected for further characterization.
Comment:
The selective isolation using MAC supplemented with cefotaxime (CTX) may introduce a selection bias by favoring the growth of cephalosporin-resistant E. coli strains. Please clarify whether this approach was intentionally designed to screen specifically for extended-spectrum β-lactamase (ESBL) or cephalosporin-resistant strains. If so, this should be clearly stated and justified in the methods section. Additionally, explain whether E. coli isolates from non-supplemented MAC plates were also included for comparative purposes.
2. Results (Lines 194–197):
A total of 111 E. coli isolates were recovered from MAC + CTX plates, distributed among three poultry suppliers.
Comment:
The current description may further support concerns regarding selection bias. Please elaborate on the rationale for using only cefotaxime-supplemented MAC for isolation and whether this might have influenced the diversity or representativeness of the E. coli population studied.
3. Figures 1–3:
Comment:
Please ensure that all axes in Figures 1 through 3 are clearly labeled, including units where applicable. Both X- and Y-axes must be fully defined in the figure legends to enhance interpretability for the readers.
4. Table 1 – Presentation of Isolate Characteristics:
Comment:
The current format of Table 1, which presents detailed individual characteristics of all 111 E. coli isolates, is overly complex and may overwhelm the reader. Consider restructuring the data into summarized categories—such as resistance profiles, virulence gene patterns, or clustering by supplier or resistance class. Alternatively, present only the key summary statistics in the main table and move the detailed isolate data to a supplementary table.
Author Response
Thank you for your valuable comments of our manuscript entitled “Phenotypic and Molecular Study of Multidrug-Resistant Escherichia coli Isolates Expressing Diverse Resistance and Virulence Genes from Broilers in Tunisia” Manuscript Number.: antibiotics-3799029. Please find attached our responses to the comments from the Reviewers. We have incorporated the necessary changes in the manuscript, which are highlighted in yellow.
Best regards,
Dr. Ghassan Tayh
Point #1:
- Materials and Methods (Lines 130–136):
The methodology describes the isolation of E. colifrom caecal content using MacConkey agar (MAC) and MAC supplemented with cefotaxime (1 μg/ml). A single colony was selected for further characterization.
The selective isolation using MAC supplemented with cefotaxime (CTX) may introduce a selection bias by favoring the growth of cephalosporin-resistant E. coli strains. Please clarify whether this approach was intentionally designed to screen specifically for extended-spectrum β-lactamase (ESBL) or cephalosporin-resistant strains. If so, this should be clearly stated and justified in the methods section. Additionally, explain whether E. coli isolates from non-supplemented MAC plates were also included for comparative purposes.
Response: About the protocol of bacterial isolation using MAC supplemented with cefotaxime (CTX), some references used this protocol such as: Evaluation and validation of laboratory procedures for the surveillance of ESBL-, AmpC-, and carbapenemase-producing Escherichia coli from fresh meat and caecal samples. We cite the reference in the revised manuscript (Please see Page No. 13 in line 432). Furthermore, we picked up a colony from MAC supplemented with CTX culture for each sample, whereas the E. coli isolates from non-supplemented MAC plates were included for prevalence rate purpose (Please see Page No. 13 in lines 433-436).
Point #2: Results (Lines 194–197):
A total of 111 E. coli isolates were recovered from MAC + CTX plates, distributed among three poultry suppliers.
Comment:
The current description may further support concerns regarding selection bias. Please elaborate on the rationale for using only cefotaxime-supplemented MAC for isolation and whether this might have influenced the diversity or representativeness of the E. coli population studied.
Response:
Regarding the use of MAC supplemented with 1 mg/L cefotaxime (CTX) for detecting broad-spectrum cephalosporins resistance and ESBL-producing E. coli.
Extended-spectrum β-lactamase- (ESBL) producing Enterobacterales are widely distributed and emerging in both human and animal reservoirs worldwide. These CTX resistance strains were used for further molecular investigation such as resistance genes, virulence, integrons and phylogenetic typing. On the other handthe E. coli isolates from non-supplemented MAC plates were included for prevalence rate purpose.
In our study the frequence rate in MAC with and without CTX was 100%.
Point #3: Figures 1–3:
Comment:
Please ensure that all axes in Figures 1 through 3 are clearly labeled, including units where applicable. Both X- and Y-axes must be fully defined in the figure legends to enhance interpretability for the readers.
Response: We agree with the reviewer and the correction of the three figures has been made in the revised manuscript.
Point #4: 4. Table 1 – Presentation of Isolate Characteristics:
Comment:
The current format of Table 1, which presents detailed individual characteristics of all 111 E. coli isolates, is overly complex and may overwhelm the reader. Consider restructuring the data into summarized categories—such as resistance profiles, virulence gene patterns, or clustering by supplier or resistance class. Alternatively, present only the key summary statistics in the main table and move the detailed isolate data to a supplementary table.
Response: Thank you for your comment. We have added a new table summarizing the frequency of the detected genes, and the original detailed isolate data (table 1) have been moved to the Supplementary File, placed after the references. (Please see Page No. 5 in lines 152-154).

Reviewer 3 Report
Comments and Suggestions for Authors
Dear Authors,
I appreciate your efforts in researching this important topic. However, the manuscript requires significant revision to improve its scientific clarity, structure, and language. Please address the points raised in the review carefully and thoroughly.
Best wishes for your continued work.
- Is there a typographical error in the abstract where it says “fequently” instead of “frequently”? Please correct the spelling for clarity and professionalism.
- What does “Several E. coli were isolated” specifically mean? How many isolates were taken per caecal sample? The wording is vague. Please clarify the exact number of isolates per sample.
- What is the source for the claim that “by 2050, antimicrobial resistance will become the leading cause of death”? Such a strong statement requires citation. Please include the exact source.
- Can you streamline the introduction to reduce overlap when describing ExPEC and APEC? Some repetition exists; try to merge or clarify overlapping concepts.
- Why were only three suppliers chosen, and only during the first four months of 2022? Please explain the rationale behind this time frame and sample source limitation.
- Does selective culturing on MacConkey agar with cefotaxime introduce bias by excluding non-resistant E. coli? This might inflate resistance estimates. Was total E. coli also isolated for comparison?
- Can you include a summary table of primers with annealing temperatures and amplicon sizes in the main text?
- How did you confirm the assumptions for using Pearson’s Chi-square test were met (e.g. minimum expected counts)?
- Enhance figure clarity by adding error bars, Figure 1
- Could Table 1 be split into multiple smaller tables to improve readability?
- How do you explain the detection of blaIMP in broiler E. coli isolates, given its rarity? Please discuss possible environmental or management-related sources.
- Are there more recent studies (2021–2024) for ESBL prevalence comparisons instead of older references? Update the discussion to reflect the latest data and trends.
- Have you validated your suggestion of horizontal gene transfer with conjugation assays or plasmid profiling?
- Why is there no paragraph discussing study limitations such as sampling bias or lack of genomic context?
- Have all abbreviations (e.g., MDR, ESBL, APEC) been defined upon first use in the text? Please ensure consistency in abbreviation usage.
- Is there any information on antimicrobial use practices at the sampled poultry farms?
- Have you considered extending phylogenetic analysis beyond Clermont typing, such as MLST?
- This could help in tracking clonal lineages and resistance dissemination.
The article needs English proofreading.
Author Response
Thank you for your valuable comments of our manuscript entitled “Phenotypic and Molecular Study of Multidrug-Resistant Escherichia coli Isolates Expressing Diverse Resistance and Virulence Genes from Broilers in Tunisia” Manuscript Number.: antibiotics-3799029. Please find attached our responses to the comments from the Reviewers. We have incorporated the necessary changes in the manuscript, which are highlighted in yellow.
Best regards,
Dr. Ghassan Tayh
Point #1: Is there a typographical error in the abstract where it says “fequently” instead of “frequently”? Please correct the spelling for clarity and professionalism.
Response: Done.
Point #2: What does “Several E. coli were isolated” specifically mean? How many isolates were taken per caecal sample? The wording is vague. Please clarify the exact number of isolates per sample.
Response: We agree with the reviewer and the correction has been made in the revised manuscript: (Please see Page No. 13 in lines 433-436).
Point #3: What is the source for the claim that “by 2050, antimicrobial resistance will become the leading cause of death”? Such a strong statement requires citation. Please include the exact source.
Response: Regarding the reference of the raise of death caused by AMR in 2050, the reference has been added in the revised manuscript: (Please see Page No. 2 in line 80-83).
Point #4: Can you streamline the introduction to reduce overlap when describing ExPEC and APEC? Some repetition exists; try to merge or clarify overlapping concepts.
Response: As suggested by the reviewer the corrections have been made in the revised manuscript: (Please see Page No. 2 in lines 64-76).
Point #5: Why were only three suppliers chosen, and only during the first four months of 2022? Please explain the rationale behind this time frame and sample source limitation.
Response: The three selected suppliers are the main poultry meat providers in Tunisia, each located in one of the country’s three major regions: Greater Tunis (the capital), Sousse (central-east Tunisia), and Sfax (southeast Tunisia). The sampling period from January to April 2022 was chosen to cover the winter and spring seasons (in Tunisia spring begins very early) , allowing us to examine the frequency of bacterial isolates in chickens during these periods. In practice, the time frame was also determined by the overall schedule of the multi-country project, which coordinated laboratory activities across five participating nations.
Samples were collected from the slaughterhouses of these suppliers to assess bacterial isolates and their antimicrobial resistance in chicken meat intended for human consumption.
Point #6: Does selective culturing on MacConkey agar with cefotaxime introduce bias by excluding non-resistant E. coli? This might inflate resistance estimates. Was total E. coli also isolated for comparison?
Response: Regarding the use of MAC supplemented with 1 mg/L cefotaxime (CTX) for detecting broad-spectrum cephalosporins resistance and ESBL-producing E. coli.
Extended-spectrum β-lactamase- (ESBL) producing Enterobacterales are widely distributed and emerging in both human and animal reservoirs worldwide. These CTX resistance strains used for further molecular investigation such as resistance genes, virulence, integrons and phylogenetic typing. In the other hand the E. coli isolates from non-supplemented MAC plates were included for prevalence rate purpose.
In our study the frequence rate in MAC with and without CTX was 100%.
Point #7: Can you include a summary table of primers with annealing temperatures and amplicon sizes in the main text?
Response: Thank you for your comment. Regarding the primers, annealing temperatures and amplicon sizes have included in Supplementary file 1&2.
Point #8: How did you confirm the assumptions for using Pearson’s Chi-square test were met (e.g. minimum expected counts)?
Response: We used SPSS to analyze the data and determine the frequency of bacterial isolates, resistance genes, virulence factors, and phylogenetic groups. Pearson’s Chi-square test was applied to calculate the P-values for comparisons, such as the distribution of resistance genes and virulence factors across different phylogenetic groups. Details regarding the determination of bacterial isolate frequencies and other related data have been added in the revised manuscript (see Page 14, lines 491–493).
Point #9: Enhance figure clarity by adding error bars, Figure 1
Response: Done.
Point #10: Could Table 1 be split into multiple smaller tables to improve readability?
Response: Thank you for your comment. We have added a new table summarizing the frequency of the detected genes, and the original detailed isolate data (table 1) have been moved to the Supplementary File, placed after the references.
Point #11: How do you explain the detection of blaIMP in broiler E. coli isolates, given its rarity? Please discuss possible environmental or management-related sources.
Response: As suggested by the reviewer explanation has been added in the revised manuscript: (Please see Page No. 10 in line 302-306).
Point #12: Are there more recent studies (2021–2024) for ESBL prevalence comparisons instead of older references? Update the discussion to reflect the latest data and trends.
Response: As suggested by the reviewer the correction has been made in the revised manuscript: (Please see Page No. 19 in line 213-215 and 219-224).
Point #13: Have you validated your suggestion of horizontal gene transfer with conjugation assays or plasmid profiling?
Response: We agree with the reviewer that confirmation of horizontal gene transfer should be performed by using conjugation assays or plasmid profiling. Although we did not conduct these analyses, we have suggested in the conclusion that there is a possibility of horizontal transfer of these genes to bacteria in the environment and humans, which would further exacerbate concerns regarding antibiotic resistance.
Point #14: Why is there no paragraph discussing study limitations such as sampling bias or lack of genomic context?
Response: As suggested by the reviewer, a paragraph discussing study limitations has been added in the revised manuscript: (Please see Page No. 12 in line 410-415).
Point #15: Have all abbreviations (e.g., MDR, ESBL, APEC) been defined upon first use in the text? Please ensure consistency in abbreviation usage.
Response: We have checked that all abbreviations are defined in full upon their first appearance in the text and have ensured consistency in their usage throughout the manuscript.
Point #16: Is there any information on antimicrobial use practices at the sampled poultry farms?
Response: We have added a paragraph on antimicrobial use in the poultry sector in Tunisia to the revised manuscript (see Page 9, lines 343–348). However, we do not have specific information on antimicrobial use practices at the sampled poultry farms.
Point #17: Have you considered extending phylogenetic analysis beyond Clermont typing, such as MLST?
Response: No, we did not perform MLST in this study. We acknowledge this as a limitation and have highlighted it accordingly in the manuscript.
Point #18: This could help in tracking clonal lineages and resistance dissemination.
Response: We agree with the reviewer.

Round 2
Reviewer 2 Report
Comments and Suggestions for Authors
Please adjust the column widths in Table 2 to improve readability.
Author Response
Response letter
Thank you for your valuable comments of our manuscript entitled “Phenotypic and Molecular Study of Multidrug-Resistant Escherichia coli Isolates Expressing Diverse Resistance and Virulence Genes from Broilers in Tunisia” Manuscript Number.: antibiotics-3799029. Please find attached our responses to the comments from the Reviewers. We have incorporated the necessary changes in the manuscript, which are highlighted in yellow.
Best regards,
Dr. Ghassan Tayh
Reviewer 2
Point #: Please adjust the column widths in Table 2 to improve readability.
Response: As suggested by the reviewer the correction has been made in the revised manuscript: (Please see Page No. 7 in line 194).
Reviewer 3 Report
Comments and Suggestions for Authors
Your response is looking ok
Author Response
Thank you for your effort.